# Unraveling the Transcriptional Dynamics of NASH Pathogenesis Affecting Atherosclerosis

**DOI:** 10.3390/ijms23158229

**Published:** 2022-07-26

**Authors:** Anita M. van den Hoek, Serdar Özsezen, Martien P. M. Caspers, Arianne van Koppen, Roeland Hanemaaijer, Lars Verschuren

**Affiliations:** 1Department of Metabolic Health Research, The Netherlands Organization for Applied Scientific Research (TNO), 2333 CK Leiden, The Netherlands; arianne.vankoppen@tno.nl (A.v.K.); roeland.hanemaaijer@tno.nl (R.H.); 2Department of Microbiology and Systems Biology, The Netherlands Organization for Applied Scientific Research (TNO), 3704 HE Zeist, The Netherlands; serdar.ozsezen@tno.nl (S.Ö.); martien.caspers@tno.nl (M.P.M.C.); lars.verschuren@tno.nl (L.V.)

**Keywords:** NAFLD, NASH, inflammation, metabolic syndrome, atherosclerosis, systems biology, organ cross-talk

## Abstract

The prevalence of non-alcoholic steatohepatitis (NASH) is rapidly increasing and associated with cardiovascular disease (CVD), the major cause of mortality in NASH patients. Although sharing common risk factors, the mechanisms by which NASH may directly contribute to the development to CVD remain poorly understood. The aim of this study is to gain insight into key molecular processes of NASH that drive atherosclerosis development. Thereto, a time-course study was performed in Ldlr−/−.Leiden mice fed a high-fat diet to induce NASH and atherosclerosis. The effects on NASH and atherosclerosis were assessed and transcriptome analysis was performed. Ldlr−/−.Leiden mice developed obesity, hyperlipidemia and insulin resistance, with steatosis and hepatic inflammation preceding atherosclerosis development. Transcriptome analysis revealed a time-dependent increase in pathways related to NASH and fibrosis followed by an increase in pro-atherogenic processes in the aorta. Gene regulatory network analysis identified specific liver regulators related to lipid metabolism (SC5D, LCAT and HMGCR), inflammation (IL1A) and fibrosis (PDGF, COL3A1), linked to a set of aorta target genes related to vascular inflammation (TNFA) and atherosclerosis signaling (CCL2 and FDFT1). The present study reveals pathogenic liver processes that precede atherosclerosis development and identifies hepatic key regulators driving the atherogenic pathways and regulators in the aorta.

## 1. Introduction

Non-alcoholic fatty liver disease (NAFLD) is closely associated with obesity, insulin resistance and dyslipidemia and is considered to be the hepatic manifestation of the metabolic syndrome. NAFLD entails a spectrum of liver diseases that range from simple steatosis to non-alcoholic steatohepatitis (NASH), which can result in cirrhosis and ultimately hepatocellular carcinoma and organ failure. Due to the worldwide epidemic of obesity, the prevalence of both NAFLD and NASH continue to rise and this disease has become the most common cause of liver dysfunction. The presence and severity of NAFLD is related to increased mortality, and liver fibrosis in particular has been demonstrated to be a strong predictor for NAFLD-related mortality [1,2]. Although this increased mortality can be due to adverse hepatic outcomes, the primary cause of mortality in patients with NASH remains cardiovascular disease (CVD) [3]. The association of NAFLD with the incidence of CVD has been numerously described and many studies using cardiovascular risk assessment tools have reported increased cardiovascular risks in NAFLD patients ([4,5,6] and references herein).

Despite the strong association of NAFLD with CVD, the precise mechanisms linking NAFLD to cardiovascular complications remain elusive and it is still unknown whether NAFLD can be an independent driver of CVD. Several pathophysiological mechanisms can play a pivotal role in the link between NAFLD and CVD, such as alterations in lipid metabolism, insulin resistance, inflammation and oxidative stress [7,8]. Due to an increased overflow of free fatty acids (FFAs) to the liver and an increased lipogenesis in NASH patients, the lipid profile in these patients shifts towards increased levels of triglycerides and low-density lipoprotein (LDL)- and very low-density lipoprotein (VLDL)-cholesterol, all known cardiovascular risk factors [9,10,11,12]. The increased overflow of FFAs to the liver may lead as well to lipotoxicity and oxidative stress [13]. Oxidative stress induces endothelial dysfunction and thereby enhances the atherosclerotic process and predisposes NAFLD subjects to develop cardiovascular events [14]. Furthermore, hepatic insulin resistance is considered a key pathophysiological mechanism of NAFLD [15] and leads to hyperglycemia that can subsequently accelerate atherosclerosis [16,17]. In addition, hepatic inflammation, one of the hallmarks of NASH, leads to the secretion of proinflammatory cytokines and chemokines and several of those inflammatory markers (interleukins 1 and 6, c-reactive protein and tumor-necrosis factor-α) increase cardiovascular risks as well [18]. Due to all these shared pathophysiological mechanisms between NAFLD and CVD, the cause/consequence relationship is difficult to separate since it remains challenging to investigate the direct contribution of NAFLD to CVD.

The aim of the current study is to search for the missing link between NAFLD and CVD and to gain more insight into the dynamics of key molecular processes of NASH that drive atherosclerosis development. To this end, a novel in silico approach (dynamical GENIE3) was used to identify hepatic regulators that are connected to regulators in the aorta. For this in silico approach, the gene expression data of a time-course study in Ldlr−/−.Leiden mice was used. Ldlr−/−.Leiden mice are genetically predisposed to develop CVD and these mice have been extensively characterized for recapitulating features of metabolic syndrome and NASH when fed a high-fat diet [19,20,21,22,23,24,25,26,27,28] and are one of the few animal models that develop both NASH and CVD when fed a translational diet. NASH and atherosclerosis development were histologically analyzed after 6, 12, 18 and 24 weeks of high-fat diet feeding. Transcriptome analysis of liver and aorta was subsequently performed and revealed a time-dependent increase in pathways related to NASH and fibrosis followed by an increase in pro-atherogenic processes in the aorta. Using the differentially expressed genes (DEGs) from the time series, a gene regulatory network analysis (dynamical GENIE3) was performed. This way, we identified specific liver regulators related to lipid metabolism, inflammation and fibrosis, which are linked to a set of aorta target genes related to vascular inflammation and atherosclerosis signaling. Verification of these identified liver regulators was performed in an independent second study.

## 2. Results

### 2.1. HFD Feeding Induces Features of the Metabolic Syndrome, NASH and Atherosclerosis

Ldlr−/−.Leiden mice fed a high-fat diet (HFD) for 24 weeks developed pronounced obesity compared to the age-matched control mice fed a low fat chow diet (Table 1). Plasma insulin levels were significantly increased on the HFD diet as compared to the chow diet at t = 24 (10.6-fold, *p* < 0.001), while glucose levels remained similar, resulting in a significantly higher insulin resistance on the HFD (10.3-fold increase in HOMA-IR at t = 24, *p* < 0.001) (Table 1). In response to the HFD, the mice developed hypercholesterolemia (4.5-fold, *p* < 0.001) and severe hypertriglyceridemia (3.8-fold, *p* < 0.001) (Table 1).

In parallel with the development of the obese phenotype and plasma characteristics of the Metabolic Syndrome, the Ldlr−/−.Leiden mice developed NASH as indicated by the rapid induction of histopathological steatosis upon HFD feeding (Table 1). A quantitative analysis revealed that after 24 weeks about 59% of the surface area was steatotic (Table 1) of which half consisted of macrovesicular steatosis and half of microvesicular steatosis (Figure 1a, see also [29]). HFD feeding also strongly induced lobular inflammation, characterized by aggregates of inflammatory cells comprising mononuclear cells and polymorphonuclear cells. Quantification of the lobular inflammation demonstrated that the HFD feeding resulted in a comparable number of aggregates after 6 and 12 weeks, and demonstrated a strong increase as compared to the chow diet after 18 and 24 weeks (resulting in a 18.1-fold increase after 24 weeks, *p* < 0.001) (Table 1). Hepatic fibrosis was not present after 6 and 12 weeks of HFD feeding, became detectable in some mice per group after 18 weeks, and increased further (and in more mice per group) after 24 weeks, although not significantly (*p* = 0.153) (Table 1, Figure 1a).

Twelve weeks of HFD feeding induced onset of atherosclerosis, as shown by the increased atherosclerotic lesion area (2.9-fold vs. chow, *p* = 0.008) and after 18 and 24 weeks the atherosclerosis was evidently existing (resulting in a 6.2-fold increase after 24 weeks, *p* < 0.001) (Figure 1a,b). Next to the lesion area, lesion severity was evaluated as well. Lesion severity significantly shifted during prolongation of the HFD towards less mild lesions (from 90% after 12 weeks to 34% after 24 weeks) and more severe lesions (from 10% after 12 weeks to 66% after 24 weeks) (Figure 1c). Furthermore, the severity of the atherosclerotic lesion area after 24 weeks revealed a significant quadratic correlation with hepatic steatosis (R^2^ = 0.77, *p* < 0.001) (Figure 1d), thereby further emphasizing the role of NASH pathogenesis within atherosclerosis development.

### 2.2. Transcriptome Analysis Showed Dynamics of Key Processes Involved in NASH and Atherosclerosis

To gain more insight into the dynamics of key molecular processes of NASH that drive atherosclerosis development, a transcriptome analysis was performed on liver and aorta samples and time-resolved patterns of regulation were analyzed. Thereto, the differentially expressed pathways (DEPs) in liver and aorta of Ldlr−/−.Leiden mice on HFD were compared to the mice on chow diet for all different time-points (t = 6, 12, 18 and 24 weeks). In the liver, HFD feeding rapidly increased the number of DEPs compared to chow feeding, while in the aorta the total number of DEPs was lower and the increase appeared later in time (Figure 2a). After 24 weeks, the HFD feeding led to a total number of 337 DEPs in liver and 91 in aorta and the majority of those DEPs in the aorta overlapped with those of the liver (66%) (Figure 2b).

More detailed analysis of the hepatic gene expression demonstrated that the majority of DEPs at t = 12 remain differentially regulated at week 18 and 24. Furthermore, the majority of pathways expressed at week 18 and 24 were shared (90% of DEPs at t = 24 were already differentially expressed at t = 18) (Supplemental Appendix A). Among the top most significantly enriched pathways in the liver after 24 weeks was the hepatic fibrosis/hepatic stellate cell activation pathway and were several lipid and inflammatory pathways (as previously described by [29]). We performed a time-resolved enrichment analysis for the liver of the top significant canonical pathways (using enrichment significance cutoff value −log (*p*-value) > 5). Subsequent integration of expression data from all time points demonstrated a time-resolved response of the main categories of processes that play a role in the development of NASH, namely lipid metabolism, inflammation and fibrosis ([29] and Figure 2c).

Time-resolved analysis of the top canonical pathways and integration of expression data of the aorta demonstrated that in the aorta, similar as for the liver, first processes of lipid metabolism were activated, followed by inflammatory processes and fibrosis processes (Figure 2c). As can be observed in Figure 2c, the responses in the liver preceded the responses in the aorta. The process of lipid metabolism was first activated in the liver from week 6 onward and in the aorta the activation started at a lower level at week 6 and peaked at week 18. For the inflammatory processes, the activation in the liver started at week 12 and in the aorta these inflammatory processes were found to be activated in week 18 and 24. Fibrotic pathways in the liver started to be upregulated at week 12 (long before the onset of fibrosis development) and increased further at week 18 and 24, while in the aorta these processes started to have a small contribution to the top canonical pathways at week 18 and increased further at week 24. In addition, the process of atherosclerosis signaling was not significantly activated yet in the aorta at t = 12 weeks (despite the slight but significantly increased atherosclerotic lesion area in the aortic root area at this time-point), but was on rank 14 of the most enriched pathways at t = 18 weeks (-log(*p*-value) = 2.7) and rank 32 at t = 24 weeks (-log(*p*-value) = 4.6). The top 20 of the most enriched pathways in the aorta after 24 weeks of HFD feeding are visualized in Figure 2d and demonstrate that these pathways primarily consist of inflammatory pathways, followed by lipid metabolism pathways all related to cholesterol biosynthesis (Figure 2d).

### 2.3. Gene Regulatory Network Analysis Identified Key Hepatic Regulators Driving Atherogenic Pathways in the Aorta

To identify hepatic key regulators of pivotal pathogenic mechanisms of NASH development that subsequently drives atherosclerosis, a gene regulatory network analysis was performed. To this end, liver pathways such as cholesterol biosynthesis, hepatic fibrosis and atherosclerosis signaling (all related to lipid metabolism or fibrosis) were selected, and differentially expressed genes from those pathways of all time points were used as input. To avoid bias in the dynGENIE3 analysis (a balanced amount of input and output genes is required), a second similar analysis was performed, now with liver pathways related to inflammation as input (acute phase response signaling and leukocyte extravasation signaling pathways) and the same output aorta target genes. Association between those hepatic genes and differentially expressed target genes in the aorta were calculated using dynGENIE3 and key regulators and pathways in liver (regulator) and aorta (target) were identified (Figure 3a). As expected for the first part of the analysis, several key regulators in the liver were related to lipid metabolism, such as LCAT (lecithin-cholesterol acyltransferase, involved in the conversion of cholesterol into cholesteryl ester and lipoprotein assembly), ACAT2 (acetyl-Coenzyme A acetyltransferase 2, involved in absorption of cholesterol and lipoprotein assembly), SC5D (cholesterol biosynthesis sterol C5 desaturase) as well as HMGCR (HMG-CoA reductase), SQLE (squalene monooxygenase) and MVK (Mevalonate kinase) all key enzymes in cholesterol biosynthesis. Despite, or perhaps due to (as a counteracting response), the high plasma cholesterol levels found in the mice, all regulators related to cholesterol biosynthesis were found to be down-regulated in the liver upon HFD feeding. In addition, several hepatic key regulators were identified that were related to inflammation, such as IL1A (interleukin 1α) and VCAM1 (vascular adhesion molecule 1), which are both pro-inflammatory factors that were found to be up-regulated upon HFD feeding. In addition, NFKB2 was found to be upregulated, the p100 subunit of nuclear factor NF-kappa-B, a key regulator of inflammatory responses in the liver that drives the regulation of many genes reported to play a role in atherosclerosis development [30]. Furthermore, an interesting key regulator such as PDGFD (platelet-derived growth factor D), an important regulator and therapeutic target for hepatic fibrosis [31], was found to be upregulated as well.

These hepatic key regulators were associated with several key target genes in the aorta, as can be observed in Figure 3a. Most of these key targets in the aorta were found to have a role in inflammation. Several regulators are part of the innate immune system, such as CCL2, CCL7, CCL9, CCL19, CCL25, and CXCL16, which all belong to the CC and CXC chemokine family that attract monocytes, leukocytes, natural killer cells and other immune cells to the site of inflammation. In addition, the pro-inflammatory cytokine TNF-α (tumor necrosis factor α) is one of the aortic key regulators, as well as CD86 (cluster of differentiation 86), a glycoprotein expressed on macrophages and dendritic cells that involves T cell activation and is a critical driver in the pathogenesis of atherosclerosis. Furthermore, FCERG1 (IgE receptor 1), a regulatory player involving in initiating the transfer of T-cells to T-helper cells, was found as a key regulator. In addition, TNFRSF13B (tumor necrosis factor receptor superfamily member 13B; a transmembrane protein of the TNF receptor superfamily), as well as H2-Eb2 (histocompatibility 2, class II antigen E beta2; involved in antigen processing and presentation of peptide via MHC class II) all somehow involved in the innate or adaptive immune system. These inflammatory aortic key target genes could be linked to the processes of atherosclerosis signaling, (a)granulocyte adhesion and diapedesis and communication between innate and adaptive immune cells. In addition, the process cholesterol biosynthesis in the aorta was linked via the hepatic key regulators of cholesterol biosynthesis in the liver. Where the aortic target genes, DHCR7 (7-dehycrocholesterol reductase), FDFT1 (farnesyl diphosphate farnesyl transferase 1), HMGCS2 (3-hydroxy-3-methylglutaryl-CoA synthase 2), PMVK (Phosphomevalonate kinase), are all catalyzing the production of cholesterol.

The identification of the hepatic key regulators was subsequently verified by a second independent mouse study with Ldlr−/−.Leiden mice fed the high-fat diet (HFD) for 30 weeks, followed by a lifestyle intervention (switch to healthy chow diet combined with voluntary exercise) for another 20 weeks. In this study, mice on the HFD diet developed severe NASH and hepatic fibrosis together with atherosclerosis and the lifestyle intervention reversed NASH/fibrosis and had beneficial effects on atherosclerosis development as well [32]. All hepatic key regulators that were identified, except one (APOC4), were differentially expressed in this study as well upon HFD feeding and all except one (C1R) were up- or down-regulated in the same direction (Figure 3b). APOC4 was the only key regulator that was not differentially expressed in this second intervention study, while being down-regulated (2logR = −1.021, *p* < 0.001) in our first time-course study. C1R or complement component 1r was slightly up-regulated in the second study, while being slightly down-regulated (2logR = −0.315) in our first time-course study. The lifestyle intervention that improved both NASH and atherosclerosis, led to an opposite gene expression for all hepatic key regulators (Figure 3b), except C1R (lifestyle slightly attenuated the expression) and APCS (serum amyloid P component, lifestyle had no effect), thereby strengthening our postulate that these hepatic key regulators of NASH development drive the aortic key regulators and development of atherosclerosis. In addition, for two of the hepatic key regulators identified, Col3a1 and FGG, we confirmed the up- (Col3a1) or down (FGG) regulated gene expression for those genes upon HFD feeding on the protein level (Appendix A).

## 3. Discussion

In this study, in Ldlr−/−.Leiden mice, we searched for the missing link between NAFLD and CVD and aimed to gain insight into the dynamics of key molecular processes of NASH that drive atherosclerosis development. Using transcriptome analysis in a time-course study in Ldlr−/−.Leiden mice, we demonstrated the sequence of key processes involved in the development of NASH and atherosclerosis and observed that in liver as well as aorta, the first processes of lipid metabolism were activated, followed by inflammatory processes and fibrosis processes. Furthermore, we found that the responses in the liver preceded the responses in the aorta. Via the use of a novel in silico gene regulatory network analysis approach (dynGENIE3), we identified specific liver regulators related to lipid metabolism (such as SC5D, LCAT and HMGCR), inflammation (IL1A) and fibrosis (PDGF and COL3A1), linked to a set of aorta target genes related to vascular inflammation and atherosclerosis signaling.

A time course study in animals can be most helpful to provide more mechanistic insights, especially on early processes contributing to disease development, that are often difficult to investigate in humans. However, for such analysis the use of a translational model is a prerequisite. In our study, we used Ldlr−/−.Leiden mice, a specific substrain of conventional Ldlr-/- mice with a mixed 94% C57BL/6J and 6% 129S1/SvImJ background, and one of the few mouse models that develops both NASH and atherosclerosis when fed a high-fat diet (with a macronutrient composition akin to human diets and not supplemented with supraphysiological amounts of cholesterol). Ldlr−/− mice are hyperlipidemic with a lipoprotein profile resembling humans with elevated VLDL- and primarily LDL-cholesterol instead of elevated HDL-cholesterol levels and therefore are also capable of developing atherosclerosis (in contrast to wild-type mice and many other mouse models). In contrast to wild-type mice, Ldlr−/−.Leiden mice when fed a HFD without cholesterol supplementation also develop a severe form of steatosis (macrovesicular and microvesicular), severe hepatic inflammation as well as hepatic fibrosis, as previously described [28]. We realize that each animal model has its limitations, and it is unlikely that any model can recapitulate all pathogenic processes leading to NASH and atherosclerosis in humans. Obviously, the Ldlr−/−.Leiden mice will not be able to reflect processes that rely on an intact Ldlr receptor function. To determine the translational value of our model, we previously analyzed to which extent the differentially expressed pathways distinguishing human NASH patients from normal control subjects are recapitulated in the Ldlr−/−.Leiden mice on a HFD and found, in contrast to many other diet-induced and genetic mouse models [33], a high overlap (73%) [28]. Among the pathways that were represented were important pathways such as hepatic fibrosis/stellate cell activation, several inflammatory pathways, mitochondrial dysfunction and also atherosclerosis signaling pathways. Translationality of the model was further verified on molecular level by both transcriptomics and metabolomics and in head-to-head comparison with NASH patients [24,34]. We therefore expect the model can be a valuable asset to study the disease processes comprehensively, and information from the current study can contribute to the general understanding on NASH etiology and the link to atherosclerosis.

We identified the temporal dynamics of key processes involved in the development of NASH and atherosclerosis, namely lipid metabolism, inflammatory processes and fibrosis. These temporal dynamics were supported by the histopathological observations that revealed a rapid and significant induction of steatosis after 6 weeks of HFD feeding and a strong increase in hepatic inflammation on the HFD as compared to the chow diet after 18 and 24 weeks. The onset of hepatic fibrosis became detectable after 18 weeks, and increased further after 24 weeks, but was not significantly different from the chow fed animals in the current experiment after 24 weeks. We know from previous experiments that a slightly longer HFD duration (t = 28 or t = 30 weeks) leads to a significant induction of hepatic fibrosis [28,29]. Clearly, the profibrotic processes leading to the synthesis of new extracellular matrix and deposition of collagen has started in the present study on a transcriptome level, long before the clinical symptoms manifest. For all three processes, lipid metabolism, inflammation and fibrosis, the onset of processes on transcriptome level precedes the clinical characteristics and the duration before clinical symptoms manifest can be different for each characteristic, depending on the pace these processes are accomplished. For the cardiovascular processes, the current study used the aortic root area to determine the severity of atherosclerosis development, while the aortic arch was used for transcriptome analysis. The latter area is, however, less prone to develop atherosclerotic plaques than the aortic root area due to regional differences in hemodynamic forces. We indeed already observed after 12 weeks of HFD feeding a small but significant increase in atherosclerotic plaque area in the aortic root, that further increased after 18 and 24 weeks of HFD feeding, while in the aorta the process of atherosclerosis signaling was not significantly activated yet at t = 12, but was among the significantly enriched pathways at t = 18 and t = 24 weeks.

Gene regulatory network analysis (by dynGENIE3) was used as an unbiased approach to identify hepatic regulators related to lipid metabolism, inflammation and fibrosis, linked to a set of aorta target genes related to vascular inflammation and atherosclerosis signaling. This method is using the temporal dynamics, impartial of up- or down-regulation, to link hepatic regulators to aorta genes with a similar temporal gene expression pattern. Using this method, we identified several hepatic regulators related to lipid metabolism (SC5D, LCAT and HMGCR), inflammation (IL1A) and fibrosis (PDGF and COL3A1) that were linked to a set of aorta target genes related to vascular inflammation (TNFA) and atherosclerosis signaling (CCL2 and FDFT1). An important notion to this method is that the hepatic regulators are associated to the aortic regulators via their similar, and simultaneously occurring, temporal gene expression patterns. This association does not imply that there is also a causal relationship, nor does it link the hepatic regulators with aortic regulators that have a dynamic gene expression pattern that is merely delayed in time. For further investigation whether there is also a causal relationship between the hepatic and aortic key regulators, it would be very interesting to evaluate interventions that specifically target certain hepatic key regulators and subsequently analyze whether these interventions affect the linked aortic key regulators and atherosclerosis as well. This evaluation is, however, complicated since it would require the use of interventions (in a model that develops both NASH and CVD) that are very specific in targeting the hepatic key regulator without affecting any other targets as well. Specific knock-out or knock-in models, however, do suggest a certain causal relationship. We checked literature for studies that are specifically targeting one of our hepatic key regulators and measured atherosclerosis using an appropriate model that is able to develop atherosclerosis (most mouse models contain their plasma cholesterol primarily in HDL-particles instead of VLDL/LDL particles and therefore do not develop atherosclerosis) and found a few examples. Absence of LCAT for instance, one of the hepatic key regulators that in our study was found to be down-regulated on high-fat diet and linked to several aortic key regulators, accelerates atherosclerosis in hamsters [35,36]. Similarly, SELP (selectin-P) deficiency, one of the hepatic key regulators that in our study was found to be upregulated and linked to atherosclerosis signaling via TNFSF11, reduces atherosclerosis in combined P-selectin/ApoE knock-out mice vs. ApoE single knock-out mice [37]. In addition, knock-in of human ApoA2 in rabbits reduced atherosclerosis, in line with our observation of a hepatic key regulator being down-regulated on the high-fat diet [38,39]. The data on our hepatic key regulators, identified via a gene regulatory network analysis to be linked to a set of aortic key regulators, therefore corroborate with the existing literature on knock-out and deficient models.

In summary, our results demonstrate time-resolved regulation of key processes involved in the development of NASH and atherosclerosis in HFD-fed Ldlr−/−.Leiden mice. We have identified key hepatic regulators related to lipid metabolism, inflammation and fibrosis, that were linked to a set of aorta target genes related to vascular inflammation and atherosclerosis signaling. These data have translational value and may provide interesting targets for new NASH therapeutics designed to have a beneficial effect on cardiovascular disease as well.

## 4. Materials and Methods

### 4.1. Animals and Experimental Design

All animal care and experimental procedures were approved by the Ethical Committee on Animal Care and Experimentation (Zeist, The Netherlands; approval reference numbers TNO-3553, Date: 6 February 2014 and TNO-312, Date 2 October 2017), and were in compliance with European Community specifications regarding the use of laboratory animals. Male Ldlr−/−.Leiden mice (TNO, Metabolic Health Research, Leiden, The Netherlands) were used. This substrain of the Ldlr-/- mouse has a 94% C57BL/6J background and 6% 129S1/SvImJ background. Mice were group housed in a temperature-controlled room on a 12 h light-dark cycle and had free access to food and heat sterilized water.

For the time-course experiment, 12-week-old mice were matched on age, body weight, blood glucose, plasma cholesterol and triglycerides into two groups of mice: mice that were kept on the healthy grain-based chow diet (R/M-H, Ssniff Spezialdieten GmbH, Soest, Germany) and mice that were given a high-fat diet (HFD) containing 45 kcal% fat from lard, 35 kcal% from carbohydrates (primarily sucrose) and 20 kcal% casein (D12451, Research Diets, new Brunswick, NJ, USA). The animal study was described previously in more detail [29]. In short, every six weeks a subset of mice (*n* = 6 chow and *n* = 15 HFD) was sacrificed unfasted by gradual-fill CO_2_ asphyxiation. Body weight and food intake per cage were measured regularly during the study. Blood samples were taken from the tail vain after 5 h fasting (with blood withdrawn around 08.00 h) in EDTA-coated tubes (Sarstedt, Nümbrecht, Germany). Terminal blood was collected through cardiac heart puncture to prepare EDTA plasma and livers were collected, weighed and fixed in formalin and paraffin-embedded (lobus sinister medialis hepatis and lobus dexter medialis hepatis) for histological analysis or (remaining liver lobes) fresh-frozen in N_2_ and subsequently stored at −80 °C for gene expression analysis. Hearts with the aortic root area of the same mice were collected, formalin-fixed and paraffin-embedded and used for histological analysis of atherosclerosis development. The aortic arch was collected, fresh-frozen in N_2_ and subsequently stored at −80 °C for gene expression analysis.

For the intervention study, 14–16-week-old mice were matched on age, body weight, blood glucose, plasma cholesterol and triglycerides into one group of mice that were kept on the chow diet and one group of mice that was fed the HFD. The animal study was described previously in more detail [32]. For this study, the following groups were used: mice that remained on the chow diet for 50 weeks (*n* = 10, healthy reference group), mice that remained on the HFD for 50 weeks (*n* = 17, HFD control group) and mice that were first fed the HFD for 30 weeks and then for another 20 weeks were switched to the chow diet and received a running-wheel (Mouse Igloo + Fast Trac, Datesand, Bredbury, England) with unlimited access in their cage for voluntary activity (*n* = 18, combined intervention group). Mice were sacrificed unfasted by gradual-fill CO_2_ asphyxiation. Body weight and food intake per cage were measured regularly during the study. Blood samples were taken from the tail vain after 5 h fasting (with blood withdrawn around 08.00 h) in EDTA-coated tubes (Sarstedt, Nümbrecht, Germany). Terminal blood was collected through cardiac heart puncture to prepare EDTA plasma and livers were collected, weighed and fixed in formalin and paraffin-embedded (lobus sinister medialis hepatis and lobus dexter medialis hepatis) for histological analysis or (remaining liver lobes) fresh-frozen in N_2_ and subsequently stored at −80 °C for gene expression analysis. Hearts with the aortic root area of the same mice were collected, formalin-fixed and paraffin-embedded and used for histological analysis of atherosclerosis development.

### 4.2. Plasma Biochemical Analysis

Blood glucose was measured at the time of blood sampling using a hand-held glucometer (Freestyle Disectronic, Vianen, The Netherlands). Plasma cholesterol and triglycerides were determined using enzymatic assays (CHOD-PAP and GPO-PAP, respectively; Roche Diagnostics, Almere, The Netherlands). Plasma insulin was analyzed by ELISA (Mercodia AB, Uppsala, Sweden). Homeostasis model assessment (HOMA) was used to calculate relative insulin resistance (IR). Five hours fasting plasma insulin and fasting blood glucose values were used to calculate IR, as follows: IR = [insulin (ng/mL) × glucose (mM)]/22.5.

### 4.3. Histology

Liver samples (lobus sinister medialis hepatis and lobus dexter medialis hepatis) were collected (from non-fasted mice), fixed in formalin and paraffin embedded, and 3 µm sections were stained with hematoxylin and eosin (H&E) and Sirius Red. NASH was scored blindly by a board-certified pathologist in H&E stained cross sections using an adapted grading system of human NASH [40,41]. In short, the level of macrovesicular and microvesicular steatosis was determined at 40× to 100× magnification relative to the total liver area analyzed and expressed as a percentage. Inflammation was scored by counting the number of aggregates of inflammatory cells per field using a 100× magnification (view size of 4.2 mm^2^). The averages of five random non-overlapping fields were taken and values were expressed per mm^2^. The development of hepatic fibrosis was assessed by a liver pathologist using Sirius Red stained slides to quantify the percentage of perisinusoidal fibrosis (expressed as the percentage of perisinusoidal fibrosis relative to the total perisinusoidal area).

Hearts fixed in formalin and embedded in paraffin were sectioned perpendicular to the axis of the aorta. Serial cross sections (5 μm thick with intervals of 50 μm) were stained with hematoxylin–phloxine–saffron (HPS) for histological analysis. The average total lesion area per cross section was then calculated [42,43]. For determination of lesion severity, the lesions were classified into five categories according to the American Heart Association classification [44]: (0) no lesion, (I) early fatty streak, (II) regular fatty streak, (III) mild plaque, (IV) moderate plaque, and (V) severe plaque.

### 4.4. Transcriptome Analysis

Liver gene expression data of the time-course study was obtained as previously described [29], of which the dataset is accessible at the NCBI GEO database via accession number GSE109345 and of the intervention study, as previously described [32], of which the dataset is accessible at the NCBI GEO database via accession number GSE179394. Detailed methods of tissue collection and processing as well as next generation sequencing (NGS) and statistical analyses are described in [29,32].

Vascular RNA of the time-course study was extracted from the aortic arch of the same mice using RNAqueous^®^ RNA isolation kit (ThermoFisher, Waltham, MA, USA, kit#AM1912). Total RNA concentration was determined spectrophotometrically using Nanodrop 1000 (Isogen Life Science, De Meern, The Netherlands), and RNA quality was assessed using the 2100 Bioanalyzer (Agilent Technologies, Amstelveen, The Netherlands). The NEBNext Ultra Directional RNA Library Prep Kit for Illumina was used to process the samples according to the protocol “NEBNext Ultra Directional RNA Library Prep Kit for Illumina” (NEB #E7420S/L). Strand-specific messenger RNA sequencing libraries were generated and sequenced at BaseClear (Leiden, The Netherlands). The libraries were multiplexed, clustered, and sequenced on an Illumina HiSeq2500 with a single-read 50-cycle sequencing protocol, 15 million reads per sample. The genome reference and annotation file Mus_Musculus.GRCm38 was used for analysis in FastA and GTF format. The reads were aligned to the reference sequence using Tophat 2.0.14 combined with Bowtie 2.1.0, and based on the mapped read locations and the gene annotation HTSeq-count version 0.6.1p1 was used to count how often a read was mapped on the transcript region. Selected differentially expressed genes (DEGs) were used as an input for pathway and upstream regulator analysis through the Ingenuity Pathway Analysis (IPA) suite (www.ingenuity.com, accessed on 11 May 2021).

### 4.5. DynGENIE3

GENIE3 [45] is a method for inferring gene regulatory networks from gene expression data. In brief, it trains random forest models predicting the expression of each gene in the target data set by using the expression of the expression data of source genes as input. In our study, the source dataset is the gene expression of liver samples and the target data is the gene expression of aorta samples (of the same mice using *n* = 15 mice for HFD and *n* = 6 for chow diet on all time-points). The different models are then used to derive weights for the source genes, measuring their respective relevance for the prediction of the expression of each target gene. Since our study uses data from a time-resolved analysis, we have identified regulators from source dataset (liver) for each target gene by using the dynamical GENIE3 (dynGENIE3) algorithm. DynGENIE3 is an adaptation of the GENIE3 method for gene regulatory network (GRN) inference that handles time series. In our work, we have identified regulators for each target gene by GRN inference for a selection of genes linked to predefined pathways and used the dynGENIE3 algorithm from the time-course gene expression data in liver and aorta. DynGENIE3 was used in python and documentation of the scripts can be found in github (https://github.com/vahuynh/dynGENIE3, accessed on 15 January 2022).

The output of DynGENIE3 is a matrix that indicates the variable importance measure (VIM) of the edges of the network between the genes (liver and aorta). We only selected the VIM that was above 99th percentile of all edges in the network. Further validation was performed using VIMs for the edges that were directed form liver genes to target aortic genes.

### 4.6. Statistical Analysis

All values shown represent means ± SEM. Statistical differences between groups were determined by using the non-parametric Kruskal–Wallis followed by Mann–Whitney U test for independent samples using SPSS software. A *p*-value < 0.05 was considered statistically significant. Two-tailed *p*-values were used. In the case of transcriptome analysis, differentially expressed genes were determined using the Deseq2-pipeline [46] using a statistical cut-off of *p* < 0.01 for the comparisons HFD vs. chow per timepoint. The selected differentially expressed genes (DEGs) were used as an input for pathway analysis (*p*-value < 0.01) through Ingenuity Pathway Analysis suite (www.ingenuity.com, accessed 11 May 2021). The differentially expressed pathways (DEP) and upstream regulators were selected based on pathway enrichment (*p*-value Fisher’s exact test) and z-score for directionality. A negative z-score < −2 indicates a predicted reduction in activity based on the direction of gene expression changes of target genes, while a positive z-score >2 indicates activation.

## Figures and Tables

**Figure 1 ijms-23-08229-f001:**
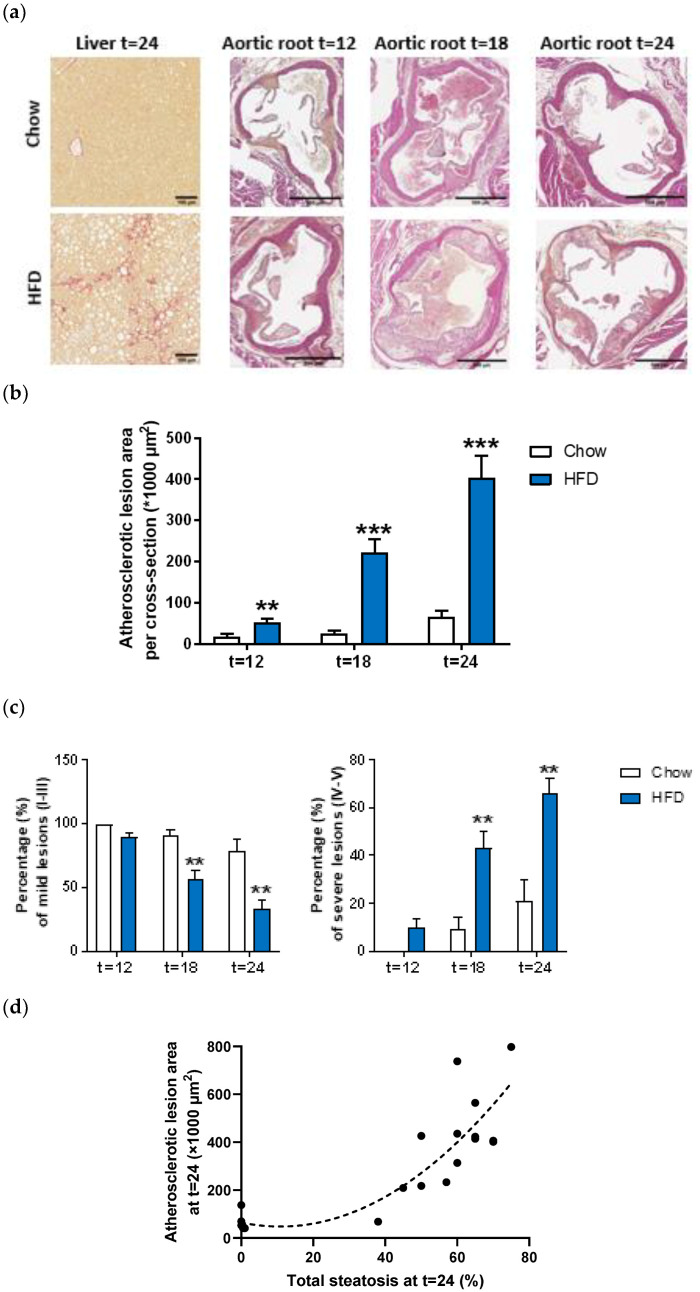
Representative images of liver cross-sections stained with Sirius red or aortic root section stained with haematoxylin-phloxine-saffron (HPS) (**a**) of Ldlr−/−.Leiden mice fed a low-fat chow diet or fed a high-fat diet (HFD) for 12, 18 or 24 weeks. Total lesion area per cross-section was quantified (**b**) and lesion severity was assessed, categorized as mild lesions (type I-III) and severe lesions (IV-V) (**c**). Furthermore, quadratic correlation of atherosclerotic lesion area with hepatic steatosis was calculated (**d**). Values represent mean ± SEM for *n* = 6 chow and *n* = 15 HFD mice/time-point. ** *p* < 0.01, *** *p* < 0.001 vs. chow.

**Figure 2 ijms-23-08229-f002:**
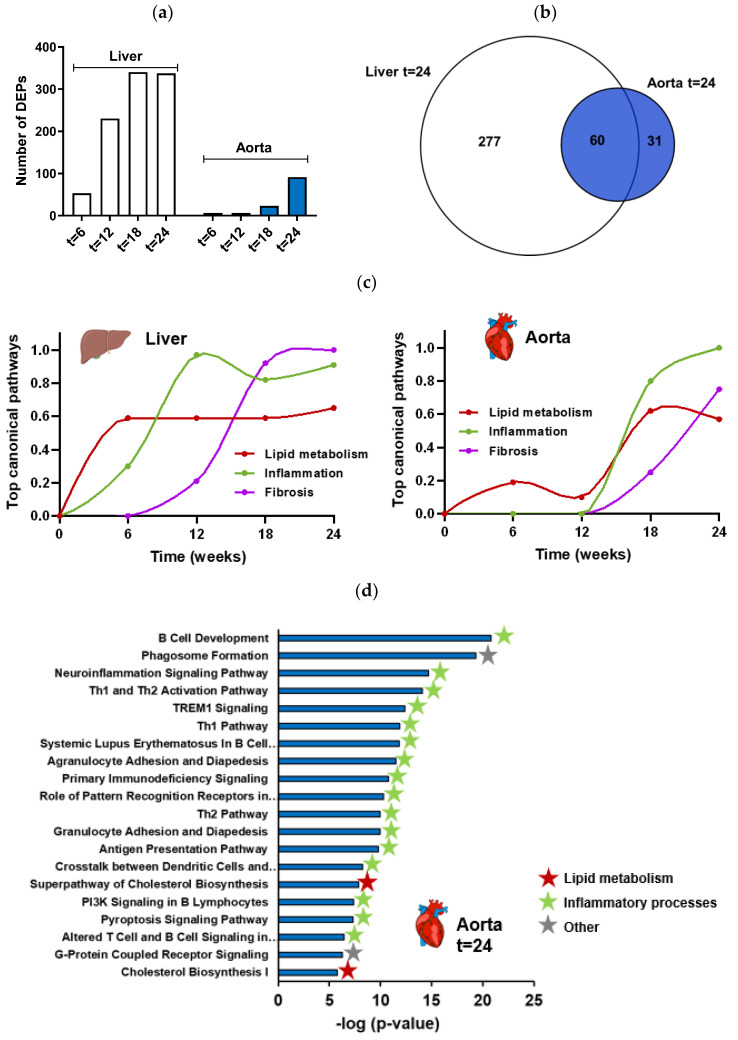
Number of differentially expressed pathways (DEPs) in liver and aorta (of the same mice) between Ldlr−/−.Leiden mice fed a high-fat diet (HFD) for 6, 12, 18 or 24 weeks (*n* = 15 per time point) vs. mice fed a healthy chow diet (*n* = 6 per time point) (**a**). Venn diagram showing the overlap of DEPs in liver and aorta in Ldlr−/−.Leiden mice after 24 weeks on HFD vs. chow diet (**b**). Graphic visualization of the temporal dynamics of key processes involved in the development of NASH (left panel) and atherosclerosis (right panel) in Ldlr−/−.Leiden mice as determined by time-resolved analysis of the top canonical pathways (**c**). The top 20 of significantly enriched biological processes (−log(*p*-value)) in aorta in Ldlr−/−.Leiden mice after 24 weeks on HFD vs. chow diet (**d**).

**Figure 3 ijms-23-08229-f003:**
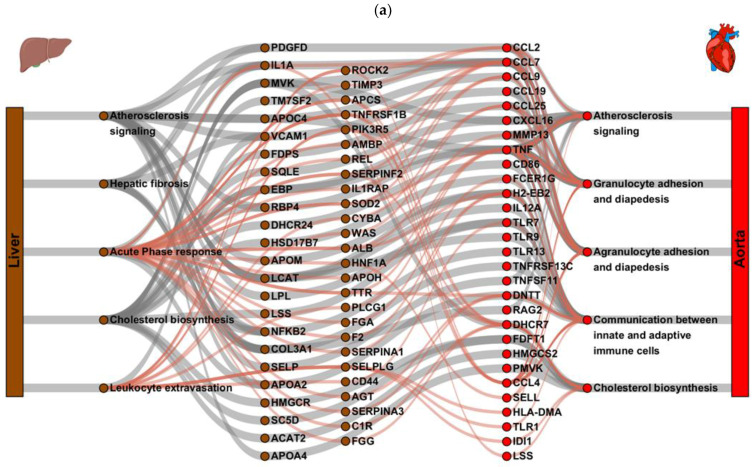
Diagram revealing the association of dynGENIE3 identified key regulators and pathways based on the differentially expressed genes of Ldlr−/−.Leiden mice fed a high-fat diet (HFD) for 6, 12, 18 and 24 weeks (*n* = 15 per time point) vs. mice fed a healthy chow diet (*n* = 6 per time point) in liver (source) and aorta (target). Two separate dynGENIE3 analyses were performed, shown here in grey and red connecting lines in one overlay figure (**a**). Hepatic key regulators were validated in a second independent study using Ldlr−/−.Leiden mice fed the high-fat diet (HFD) for 30 weeks, followed by a lifestyle intervention (switch to healthy chow diet combined with voluntary exercise) for another 20 weeks. Heatmap shows the expression of the hepatic key regulators (**b**). Cut-off values of 2logFC < −0.4 and >0.4 for HFD vs. chow were used. Blue color indicates downregulation and red color indicates upregulation.

**Table 1 ijms-23-08229-t001:** Metabolic parameters.

		Chow	HFD
Body weight (g)	t = 24	36.7 ± 1.6	51.8 ± 1.0 ***
Blood glucose (mM)	t = 24	6.7 ± 0.3	6.9 ± 0.3
Plasma insulin (ng/mL)	t = 24	2.0 ± 0.3	20.7 ± 4.3 ***
HOMA-IR	t = 24	0.6 ± 0.1	6.3 ± 1.4 ***
Plasma cholesterol (mM)	t = 24	7.6 ± 0.6	33.2 ± 2.7 ***
Plasma triglycerides (mM)	t = 24	1.3 ± 0.1	4.8 ± 0.8 ***
Total steatosis (%)	t = 6	0.0 ± 0.0	33.6 ± 3.0 ***
	t = 12	0.2 ± 0.1	53.4 ± 2.6 ***
	t = 18	1.8 ± 0.6	69.3 ± 2.7 ***
	t = 24	0.3 ± 0.2	59.3 ± 2.8 ***
Hepatic inflammation	t = 6	1.0 ± 0.4	6.1 ± 0.8 ***
(number of aggregates/mm^2^)	t = 12	2.0 ± 0.6	5.4 ± 1.2
	t = 18	2.2 ± 0.7	10.6 ± 3.0 **
	t = 24	0.7 ± 0.2	12.1 ± 3.8 ***
Hepatic fibrosis (%)	t = 6	0.0 ± 0.0	0.0 ± 0.0
	t = 12	0.0 ± 0.0	0.0 ± 0.0
	t = 18	0.0 ± 0.0	2.5 ± 2.2
	t = 24	0.0 ± 0.0	5.0 ± 2.0

Ldlr−/−.Leiden mice were fed a healthy chow diet or fed a high-fat diet (HFD) for 6, 12, 18 or 24 weeks. Values represent mean ± SEM for *n* = 6 chow and *n* = 15 HFD mice/time-point. ** *p* < 0.01, *** *p* < 0.001 vs. chow. HOMA-IR: homeostasis model assessment of insulin resistance.

## Data Availability

The mouse gene expression data set from the Ldlr−/−.Leiden time-course study described is publicly available via Gene Expression Omnibus dataset accession number GSE109345 and the dataset from the second Ldlr−/−.Leiden lifestyle intervention study that was used to validate the hepatic key regulators is available via the Gene Expression Omnibus dataset accession number GSE179394.

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
