# Peer review of "Unraveling the Transcriptional Dynamics of NASH Pathogenesis Affecting Atherosclerosis"

_ijms, 2022, doi:10.3390/ijms23158229_

Round 1

Reviewer 1 Report

This is an interesting study unravelling pathophysiological mechanisms of NAFLD and atherosclerosis through a well designed mouse model and transcriptome analysis.

However, there is no validation of the transcriptome analysis either by RT-qPCR or western plot and the findings cannot be considered as reliable. Although validation of RNA-seq methods at gene expression level (i.e.) may be questioned by some (https://www.ncbi.nlm.nih.gov/pmc/articles/PMC7823214/), however independent validation of selected upregulated genes at protein expression level (via Western blot) is well needed to show the robustness of the study findings. 

If the authors are able to provide sufficient findings of such readouts, the manuscript can be considered worthy of publication as it can provide very interesting input to the pathophysiology of NASH and atherosclerosis. 

Author Response

Response: We kindly thank you for the time and effort it took to review the paper. In our study we used transcriptome analysis via next generation sequencing. While we did not validate the gene expression data using RT-qPCR, we did validate the gene expression data using next generation sequencing of a second independent study, also with Ldlr-/-.Leiden mice fed a high fat diet. We think this approach is to be preferred above RT-qPCR validation of selected genes of the first time-course study, since it was performed using different mice and with an independent second mRNA isolation as well. As mentioned in the manuscript, lines 241-249, all hepatic key regulators that were identified, except one (APOC4), were differentially expressed in this study as well upon HFD feeding and all except one (C1R) were up- or down-regulated in the same direction. APOC4 was the only key regulator that was not differentially expressed in this second intervention study, while being down-regulated (2logR=-1.021, p<0.001) in our first time-course study. C1R or complement component 1r was slightly up-regulated in the second study, while being slightly down-regulated (2logR=-0.315) in our first time-course study. Validation on protein level was indeed not performed and we now have added the validation of a few regulators on protein level to the manuscript (Supplemental Figure 2). We previously measured proteins in this study as part of dynamic proteomics used to define a predictive molecular signature of fibrosis (as described by van Koppen et al., CMGH 2017). However, these protein analyses were performed in half of the mice per group only (n=3 chow and n=6 HFD). Two of our hepatic key regulators identified by our dynGENIE3 analysis were among those proteins measured for proteomics (Col3a1 and FGG) and the results on protein level were in line with the gene expression data. Due to the limited number of mice in the chow group, we were a bit more hesitative to add these results in the manuscript and therefore chose to add them now as supplemental data with the limited number of mice mentioned in the footnote. In response to the question of the reviewer, we tried to analyze protein levels via Western blot analysis (using all mice per group), however since we had only limited time to adapt the manuscript, we only performed this analyses for the proteins corresponding to hepatic regulators for which we had Western blot antibodies in house (NFkB and p-selectin/SELP). Of all hepatic key regulators these were the only ones we had antibodies immediately available and we performed the Western blot analyses and unfortunately this miserably failed. The antibodies might have been too old or perhaps we used the wrong dilution factor. In the end, we therefore added the Col3a1 and FGG protein data as supplemental data to the manuscript and found that for these proteins, levels were in line with the up- or down-regulation of the gene expression on HFD feeding, thereby further validating our data. The aortic gene regulators could not be verified using the second independent study or on protein level in the first time course study, since samples of aortic root/aorta were already used for other purposes (histology/transcriptomics) and therefore were not available anymore.

Reviewer 2 Report

In this study, the authors aim to find the precise mechanisms linking NAFLD and CVD. They performed pathway analysis in both liver and aorta in a time-course study in Ldlr-/- Leiden mice that fed with high fat diet, which developed NASH and CVD. They also identified key regulators in the liver that link to key genes/processes in the aorta, which may possibly drive atherosclerosis. Overall, this study is interesting, but it is too simple to fit the scope of ‘IJMS’. My major concerns with this study are listed below:

1, This study lacks novelty, the Ldlr-/- Leiden mice high fat diet model is well established and the atherosclerosis phenotype has been reported in the past. RNAseqs in this study were also done in the past.

2, Again, as mentioned in the discussion, some of the key hepatic regulators, for example, LCAT and TNFSF11, have been shown to mediate atherosclerosis in previous studies, which makes this study less novel. None of the regulators or pathways identified in this study were confirmed in mice or cells by the authors. Functional studies should be performed to strengthen this study.

Author Response

Response: We kindly thank you for the time and effort it took to review the paper. Regarding the first point, the novelty of the study, we agree with the reviewer that the Ldlr-/-.Leiden mouse model has already been described as model that develops both NASH and atherosclerosis. Indeed publications using this model with transcriptome analysis have been performed before as well and the hepatic transcriptome analysis of the first time-course study has been used previously to define a predictive molecular signature of fibrosis (as described by van Koppen et al., CMGH 2017). The novelty of our current manuscript is however based on the search for a missing link between closely associated liver and cardiovascular diseases (CVD). As also mentioned by one of the other reviewers this is actually a new initiative, and this particular topic in our view also fits with the scope of International Journal of Molecular Sciences. We used a very novel in silico approach (dynGENIE3) that is using the in vivo data of gene expression (in time) of liver and aorta to identify hepatic regulators that are connected/ associated to regulators in the aorta. With the current interest, both in literature and on international conferences, on the underlying link between NASH and CVD and the question whether NASH can be an independent driver of CVD (which was for instance a specific session topic on the recent International Conference of Fatty Liver disease of 2022), we think the research described in our manuscript fits very well to the scope of IJMS and can be a valuable contribution to this discussion. By the raised question of the reviewer, we realize however that we did not well describe that the novelty of our work is within this search for a missing link between NASH and CVD and therefore adapted the text of our manuscript accordingly to clarify this further.

Regarding the second point, lack of novelty of hepatic key regulators like LCAT and TNFSF11, we’d like to emphasize that we used an unbiased approach to unravel the pathophysiological mechanisms connecting NASH and CVD in order to identify hepatic key regulators related to lipid metabolism, inflammation and fibrosis, that are linked to a set of aorta target genes related to vascular inflammation and atherosclerosis signaling. Using this approach we identified 50 hepatic key regulators linked to 29 aortic regulators. We were therefore pleased to see that regulators like LCAT and TNFSF11 that have been shown to mediate atherosclerosis in previous studies, were indeed identified using our unbiased approach and think this strengthens the validity of our data. We think that our list of 50 hepatic key regulators provides enough interesting regulators that are novel and not mentioned yet as link between NASH and CVD in literature. Since NASH and atherosclerosis are both complex diseases with several organs playing a role (liver, WAT, muscle, gut, brain), we think that our current research question to gain more insight into the molecular processes of NASH driving atherosclerosis is rather impossible to validate using separate in vitro cell systems. We agree with the reviewer that confirmation of the regulators in mice studies would indeed be very interesting, but this research is complicated by the fact that most mouse models do not develop atherosclerosis (as most mouse models contain their plasma cholesterol in primarily HDL-particles instead of VLDL/LDL particles) and very few translational mouse models develop both NASH and atherosclerosis. Validation of the hepatic key regulators using the Ldlr-/-.Leiden model is very time-consuming and moreover very difficult since it requires interventions that are very specific for the hepatic key regulators and not affecting any other targets as well. We therefore chose to check available literature for knock-out/knock-in models with our hepatic key targets. Again, also this literature search required a background model that can develop atherosclerosis, limiting the number of studies we could find. Nevertheless, for 3 hepatic key targets we were able to find knock-out/knock-in studies that were in line with our prediction (as described in the discussion of our manuscript, lines 370-377) thereby strengthening the data of our study. We’ve now adapted our manuscript to clarify this further.

Reviewer 3 Report

van den Hoek et al. identify conserved pathways in NASH and atherosclerosis using transcriptomic analysis of liver and aorta from a unique mouse model Ldlr-/- Leiden. The authors propose that lipid metabolism, inflammation and fibrosis in NASH-induced liver can be related to aorta-specific pathways including vascular inflammation and atherogenesis. The work searching for a missing link between closely associated liver and cardiovascular diseases is a new and interesting initiative, and the topic lies in the scope of the journal. However, the data shown here are not entirely convincing so that further work to clarify the findings would be required. Below are my comments to address in the manuscript.

The limitation of the study is that the work was based on the transcriptomic data of whole tissues (liver and aorta) that comprise highly heterogeneous populations during pathogenesis. I can envision that the manuscript can be benefited from performing scRNA-seq to identify cell-specific mechanisms across the tissues. However, with time restraints, scRNA-seq is not required for this manuscript.

Next, it is not clear whether the authors used aorta and liver samples from the same mouse. In Methods, it is stated that the data of NASH-liver were used from the previously published paper. In order to understand cross-organ communications, it is important to use liver and aorta from the same mouse. The authors are required to clarify this in the manuscript to avoid confusion. In addition, there is no information on the experimental size/biological replicate in Figure 2; did you perform analysis on the data from individual mice (e.g. n=6 per tissue per timepoint) or did you combine tissue samples from 6 mice (e.g. n=1 per tissue per timepoint)? If the former are correct, have you found consistent findings from each mouse?

It is intriguing to see that Ldlr-/- Leiden mice develop NASH, atherosclerosis, obesity and diabetic conditions by HFD feeding. Despite the considerable changes in body weight, adipose tissues were not analysed/mentioned, which potentially can deliver interesting findings. If transcriptomic data analysis was performed in adipose tissues, the data comparison across those three tissues could improve the quality of the paper.

On a minor note, I noticed that the atherosclerotic lesions of Ldlr-/- Leiden mice are very small at 12 weeks of a HFD compared to widely used models such as Ldlr-/- or Apoe-/-. Since those two models do not develop NASH (if correct), is there any correlation between less progressive atherosclerosis and the development of NASH and obesity in the Ldlr-/- Leiden strain? Finally, it would be helpful to the readers if the characteristics of the Ldlr-/- Leiden mouse strain are described in comparison to other NASH and atherosclerotic mouse models.

Author Response

Response: We kindly thank you for the time and effort it took to review the paper. Regarding the first point, use of scRNA-seq, we’d like to mention that while we certainly acknowledge the added value of this technique for  many research questions, we think that in this specific case where we investigated the link between NASH and atherosclerosis, whole tissue analysis was probably the best first approach, since NASH development is also involving a change of cell population (more inflammatory cells for instance) and we were specifically interested in NASH development in the liver (so on whole tissue level, not specifically for hepatocytes) affecting atherosclerosis in the aorta. For additional research it would however indeed be interesting to investigate via scRNA-seq whether our findings are specific for a certain cell type.

Regarding the question whether liver and aorta samples were taken from the same mice and number of replicates in figure 2, we can indeed confirm that liver and aorta samples were taken from the same mice (n=6 chow and n=15 HFD mice per time-point) with consisting findings for each mouse. We for instance correlated the individual gene expression of the hepatic key regulators with the individual atherosclerotic lesion size and found a good positive correlation for regulators like LPL, VCAM1, CYBA, LPL and TIMP3 and a good negative correlation for regulators like LCAT, IL1RAP and TM7SF2, all in line with our key regulator expression (Figure 3). We indeed forgot to mention the number of mice in our manuscript in figure 2 and thank the reviewer for pointing this out. We’ve now adapted the manuscript to correct this and also added to the method section that the livers and aortas were taken from the same mice.

Regarding the Ldlr-/-.Leiden mouse model and adipose tissue, indeed the increased body weight in our mice was due to an increase in adipose tissue. We did collect different adipose tissue depots (subcutaneous, perigonadal and visceral WAT), but have not performed transcriptomics on WAT. We agree that this is interesting as well and WAT probably plays a role in NASH/CVD development as well. Cross-talk between liver, WAT and muscle and effects on NASH and atherosclerosis development has been described before in Ldlr-/-.Leiden mice on HFD (van den Hoek et al., Metabolism 2021). For our current study we focused on the link between NASH (liver) and CVD (aorta) and therefore WAT was beyond our current scope. We agree though that WAT transcriptomics is interesting as well and are considering to pursue this later in a separate manuscript.

Regarding the atherosclerotic lesion development in Ldlr-/-.Leiden mice: after 12 weeks of HFD atherosclerosis has developed, but consists of primarily mild lesions, shifting towards more severe lesions after 18 and 24 weeks on the diet. In literature, there’s already a lot of variation described for the atherosclerosis development in ApoE-/- and Ldlr-/- mice, but usually a diet is used that is supplemented with cholesterol. The cholesterol supplementation aggravates atherosclerosis development, also in our Ldlr-/-.Leiden mice, but cholesterol supplementation is not very translational (too high dose). Indeed, Ldlr-/-.Leiden mice are one of the few animal models that develops NASH and atherosclerosis when fed a high fat diet (without supplementation of diet with cholesterol, so also translational to human situation). As compared to conventional Ldlr-/- mice, the Ldlr-/-.Leiden mice are more prone to develop obesity and obesity related complications, such as insulin resistance and NASH. We think this is due to the slightly mixed background, since Ldlr-/-.Leiden mice have a 94% C57BL/6J background and 6% 129S1/SvImJ background. It has been described before that sensitivity to develop obesity related disorders is strain dependent and we think it’s rather this background difference in our mice that determines that they are more prone to develop NASH than a different atherosclerosis development. Atherosclerosis development significantly correlated in our study with steatosis (as shown in Figure 1d), but there was also a significant correlation between atherosclerosis with body weight (R2=0.67, p=0.0012), hepatic inflammation (R2=0.52, p0.017) and fibrosis (R2=0.66, p=0.0016), therefore arguing against the postulate that the less progressive atherosclerosis is linked to the development of obesity and NASH. We have now adapted the text of our manuscript to describe in more detail the characteristics of our Ldlr-/-.Leiden strain in comparison to other NASH/atherosclerosis models.